# At Risk Safety Behaviors of the Perioperative Nursing Team: A Direct Observational Study

**DOI:** 10.3390/healthcare11050698

**Published:** 2023-02-27

**Authors:** Susan Letvak, Brandi Apple, Marjorie Jenkins, Carrie Doss, Thomas P. McCoy

**Affiliations:** 1Adult Health Nursing Department, University of North Carolina, Greensboro, NC 27215, USA; 2Chapel Hill School of Nursing, University of North Carolina, Greensboro, NC 27215, USA; 3Cone Health System, Greensboro, NC 26170, USA

**Keywords:** perioperative setting, at risk safety factors, observational study

## Abstract

Background: The operating room setting has unique workforce hazards and extremely high ergonomic demands due to patient lifting/positioning requirements, long periods of standing, and the heavy equipment and supplies that are needed for surgical procedures. Despite worker safety policies, injuries among registered nurses are increasing. Most of the research on the ergonomic safety of nurses is conducted utilizing survey methodology, which may not provide accurate data. It is imperative to understand the at-risk safety behaviors that perioperative nurses face if we are to design interventions to prevent injury. Methods: Two perioperative nurses were directly observed during 60 different operating room surgical procedures (*n* = 120 different nurses). Data were collected utilizing the job safety behavioral observation process (JBSO), which is designed specifically for the operating room environment. Results: There were 82 total at-risk behaviors observed amongst the 120 perioperative nurses. More specifically, 13 (11%) of the surgical procedures had at least one perioperative nurse observed in a position of at-risk behavior, and a total of 15 (12.5%) individual perioperative nurses performed at least one at-risk behavior. Conclusion: More attention must be placed on the safety of the perioperative nurse if we are to retain a healthy, productive workforce to provide the highest quality patient care.

## 1. Introduction

The profession of nursing is known to face workplace hazards and has high ergonomic demands resulting in occupational injury. While only 61% of registered nurses (RNs) work in hospital settings, these nurses experience 74.1% of RN nonfatal workplace injuries and illnesses, with the majority of these injuries being work-related musculoskeletal injuries/disorders (MSD), which is significantly higher than the rate for all occupations [1]. Despite having worker safety policies in all hospital settings, the US Bureau of Labor Statistics [2] reports occupational injuries in registered nurses (RNs) resulting in time away from work, which increased by 42.2% from 2019 to 2020. Ergonomic-specific increases include increased contact with objects and equipment (1990 to 2830), falls/slips/trips (5320 to 5390), and over-exertion/bodily reactions (8610 to 10,510). Moreover, the state of North Carolina had a 335% increase in RN occupational injuries leading to time away from work increasing from 200 to 870 from 2019 to 2020. Workplace injuries in nurses impact not only the nurse but also have monetary and societal costs that necessitate the need to further understand the specific workplace hazards nurses face [1].

Perioperative RNs make up less than six percent of the RN workforce [3] and, thus, are often lost in the nursing workforce and ergonomic studies. However, RNs are extremely important members of the perioperative team in assuring the quality outcomes of patients undergoing surgical procedures. Shewchuk [4] states it is impossible to understand why there is a need for RNs in the operating room without first understanding the nursing roles in the operating room. One essential role is that of the circulating nurse. The circulating nurse coordinates all aspects of the perioperative environment, especially communication, and assures compliance with policies and procedures. The circulating nurse assists the entire perioperative team in maintaining a safe environment and must also be ready to step into the scrub nurse role if needed. The role of the scrub nurse, who is part of the sterile field, is to have knowledge of all steps of surgical procedures to anticipate surgeon needs by passing the needed instruments, sponges, and supplies at the correct time. A newer role for RNs in the operating room is the registered nurse first assistant (RNFA). The RNFA is an advanced practice role for nurses who perform surgical interventions under the supervision of a surgeon [5]. Specifically, the RNFA is allowed to make incisions, cauterize, suction, retract tissue, and suture surgical patients in the operating room. Finally, RNs may also have the role of certified registered nurse anesthetist (CRNA) in the operating room. The American Association of Nurse Anesthesiology [6] defines CRNA as advanced practice nurses licensed as independent practitioners who plan and deliver anesthesia, pain management, and related care to patients of all health complexities across their lifespans. In the operating room, the CRNA is responsible for the delivery of anesthesia and pain relief for the patient undergoing surgery.

The operating room setting is important for research because there are significant workforce hazards and extremely high ergonomic demands due to patient lifting/positioning requirements, long periods of standing, and the heavy equipment and supplies that are needed for surgical procedures. Moreover, incorporating ergonomic interventions into the operating room is more difficult than in other workforce settings because of demanding tasks and restrictive environments that prioritize patient safety, as well as sterility requirements [7]. It is important to note that all RNs in the operating room share safety risks, including all RN roles that are required to lift and position patients.

Because of the high ergonomic demands of the operating room, in 2011, the Association of Perioperative Registered Nurses (AORN) reviewed the research literature to identify seven ergonomic risk factors of perioperative nurses and developed ergonomic tool kits that were specific for each risk factor and updated in 2018. These tool kits provide research-supported specific safety measures for each ergonomic risk factor, such as the maximum weight to be lifted, pushed, or pulled and time limits for the use of hands as a tissue retractor and standing in one position. More recently, Yasak and Vural [8] identified additional perioperative workforce hazards, which include awkward positioning, forcible exertion, lifting while bending or twisting, overreaching, repeated motions, static postures, floors that are wet or covered with debris, heavy operating room doors, and moving oxygen tanks.

Moreover, there have been a few more recent research studies focusing on variables that influence safety hazards for perioperative nurses. Abdollahzade and colleagues [9] measured the individual demographics, job type, and working posture of 147 operating room nurses in Iran. The researchers found that gender and the type of operating room (e.g., orthopedic versus cardiac) predicted poor posture, which could lead to musculoskeletal disorders and injury. Researchers in Italy [10] analyzed the association between personal and job characteristics and the risk of upper limb work-related musculoskeletal disorders in 148 operating room nurses. Regression analyses revealed that being female and in the scrub nurse role was associated with a higher score on the disabilities of the arm, shoulder, and hand (DASH) questionnaire placing these nurses at great risk for upper limb disorders.

Most of the research conducted on the ergonomic safety of nurses was conducted utilizing survey methodology [11] and has now been outdated. Survey designs are known to have low response rates and may also have response distortion, which compromises the validity of self-report measures in high-stake situations [12], such as injury and worker safety reporting. Thus, the purpose of this study was to determine the at-risk safety factors perioperative nurses experience through direct observation in the operating room setting. It is only through understanding specific risks that interventions and needed safety education can be developed. This study was guided by applied behavioral analysis [13], which is derived from Skinner’s behavioral science theory [14] and offers a method to improve occupational safety by directly observing human behavior in the workplace setting to reduce worker injury. Observational data are useful when trying to understand situations, gather data on individual behaviors, and obtain knowledge about the physical setting [15]. Additionally, observational data can be particularly useful in the operating room setting to provide insight into this uniquely complex environment.

## 2. Methods

### 2.1. Study Design and Human Subjects Protection

We employed an observational design utilizing non-participant observation to observe the at-risk safety behaviors of perioperative nurses in the operating room. Creswell and Poth [16] defined non-participant observations as the researcher being uninvolved with the group under study: only watching and taking notes from a distance. IRB approval was first received (#21-0070). An advantage of non-participant observation is that the researcher has a more objective view of what is being observed. As part of the IRB approval, prior to observations, the participating health system required that a consent to shadow and observe form be signed by the researchers. In addition to assuring complete confidentiality of all data collected, the observers agreed that only health system perioperative nurses would be observed, and no patients would be observed.

### 2.2. Study Setting and Sample

This study was conducted in a large healthcare system located in the southeastern United States, which performs over 45,000 surgical procedures annually. The sample consisted of perioperative nurses working at the bedside in the operating room (RN circulating nurses, RN scrub nurses, CRNAs, and RNFAs). All participants had agreed to have an observer present during the surgery that was being performed.

### 2.3. Data Collection and Measurement Tool

As stated above, data collection was required following the health system’s policy of having an observer present in the operating room. The perioperative charge nurse assisted with determining suitable surgical cases which might have ergonomic risk factors to observe and obtained permission from the surgeon and perioperative team to have an observer present. Patient consent forms already included observers possibly being present. Operating room staff are used to observers being present; thus, there were no refusals or questions asked of the researcher who was present in the operating room. The researchers were careful to only observe the perioperative RNs- no observations were made of the patient, other team members, or other observers that were present during the surgical procedures. To assure as much anonymity as possible, no demographic information (such as perceived gender, age, or race/ethnicity) of the perioperative nurses observed was recorded. Additionally, no written documentation was made of any conversations that may have been overheard by the researcher.

McNaughton et al. [17] recommend that the procedures to be observed took place over a number of sites and were best observed in a range of locations. Thus, direct observations were made at the major medical center hospital, three community hospital sites, and several free-standing surgical center sites. Sixty surgical procedures, as recommended in Yasak and Vural’s [8] observational study, were observed. The surgical procedures observed included a variety of specialties to ensure capturing the ergonomic demands of RNs working in different surgical specialty lines. Observational data were collected using the job safety behavioral observation process (JBSO) designed specifically for the operating room environment by Simon et al. [18] and included all the AORN ergonomic safety risk factors. The form allowed the observer to document if the ergonomic risk factor being performed was safe (S) and followed ergonomic safety precautions or whether it was at-risk (AR) and not following ergonomic safety measures. An open-ended “comment” area was used for field notes. The co-PI, who is an experienced, certified perioperative nurse, collected observational data with a research assistant who was also highly experienced and certified in perioperative nursing. They conducted three observations together to assure inter-rater reliability with data collection.

## 3. Data Analysis

All observation checklist data were entered into SPSS.v28 software [19]. Descriptive statistics were then used to determine the specific risk factors that perioperative nurses were exposed to during surgical procedures.

### Findings

Over the course of two months, two perioperative nurses were observed during 60 different operating room surgical procedures for a total of 120 perioperative nurses who were directly observed for at-risk safety behaviors. Of the 120 different perioperative nurses observed, 60 (50%) were in the circulating nurse role, 45 (38%) were certified nurse anesthetists (CRNAs), 11 (9%) were in the scrub nurse role, and 4 (3%) were RN first assistants (RNFAs). Of the 60 surgical procedures observed, there were 82 total at-risk behaviors observed amongst the 120 perioperative nurses. More specifically, 13 (11%) of the surgical procedures had at least one perioperative nurse who was observed conducting at-risk behavior, and a total of 15 (12.5%) of the individual perioperative nurses performed at least one at-risk behavior. Table 1 identifies specific at-risk behaviors with the number and type of perioperative nurses who were at risk. The top three at-risk behaviors observed included overreaching (16 observations), working in an awkward position (15 observations), and lifting while bending or twisting (13 observations). Circulating nurses had the largest number of at-risk behaviors but also made up 50% of those observed. CRNAs had 22% of the at-risk behaviors and made up 38% of those observed. Scrub nurses had 16% of the at-risk behaviors, followed by 6% for the RNFAs; however, they made up 9% and 3% of those observed, respectively.

## 4. Discussion

Most hospitals prioritize patient safety; however, Fencl and colleagues [20] argue that patient safety also requires a just culture and prioritizing nurse safety in the perioperative work setting. This study documents that despite well-publicized AORN safety tool kits, perioperative nurses have numerous personal at-risk behaviors, as well as environmental risks, while working in the operating room setting.

First, environmental risks that place the perioperative nurse at risk of injury are heavy doors and operating room floors that have debris or are wet. Operating room doors are over-pressurized and hermetically sealed to prevent contaminated air from entering and increasing the risk of surgical infections [21]. In this study, all the operating room doors were electronic; thus, there were no observations of having to open a heavy door, especially while pushing or carrying a heavy object. Of the 60 surgical procedures observed, only one operating room had a very wet floor which exposed the CRNA and circulating nurse to a risk of falling, and five circulating nurses and two scrub nurses were at risk for falling because of debris on the floors. The United States Department of Labor, Occupational Safety and Health Administration (OSHA) [22] has a standard that walking-working surfaces must be clean and dry. When wet processes are used, dry-standing places (such as platforms or mats) must be provided, and all walking-work surfaces should be free of hazards.

The majority of the safety risks observed in this study were at the personal level. Over-reaching was observed in each of the four perioperative nurse roles (CRNA, circulating nurse, scrub nurse, and RN first assistant). Overreaching was observed when performing tasks such as placing EKG leads over a very obese patient as well as while lifting and moving a 2000 mL irrigation bag over a patient. Reaching is considered an awkward body posture that can lead to injury [23]. Surprisingly, despite the obvious strain over-reaching places on the spine, especially when reaching with weight in one’s hands, no evidence was found in the literature that specified over-reaching as an ergonomic risk factor for nurses, and “reaching” was only discussed as an awkward body position.

Fifteen of the 120 perioperative nurses were observed working in awkward positions, mostly in a twisted position while standing or hunched over. An additional 13 perioperative nurses were observed lifting patients or equipment while bending or twisting. The U.S. OSHA [24] clearly lists ergonomic risk factors for MSD injury or disorders as working in awkward positions, including twisting while lifting. Additionally, seven different perioperative nurses were observed either lifting too-heavy objects, transferring a too-heavy patient (six), positioning a too-heavy patient (six), or pushing/pulling too heavy of weight (one). These behaviors are classified by OSHA [24] as exerting excessive force, putting the worker at risk for MSD injuries and disorders.

It is important to note that 11% of the 60 surgical procedures observed had at least one perioperative nurse performing an at-risk behavior. The circulating nurse role had the largest number of at-risk behaviors (56%); however, they also made up 50% of the perioperative nurses observed. Circulating nurses, who are non-sterile members of the operating room team, are in a multi-dimensional role and are “caretakers of the highly technical surgical environment and supervisor of surgical team’s activities while concurrently directing patient care [25]. These highly trained nurses are in the critical role of assuring the safety of patients, and more efforts must be directed toward their ergonomic safety.

## 5. Implications

This observational study has implications for all who work in the perioperative setting. Each of the observed safety risk factors has the potential to put a perioperative nurse at risk for injury. Clearly, floors should be free of debris and water, and if certain surgical procedures produce excess debris and water, platforms should be provided to keep staff safe until the debris and water can be removed. Perioperative nurses must be continuously reminded of the dangers of lifting/pushing too-heavy weights and equipment over the recommended weight standards. Time pressure should not prevent perioperative nurses from seeking additional help for lifting and pushing heavy weights or obtaining the support needed for lifting equipment. Nurse managers must advocate for more time between operating room procedures to remove the time pressure placed on perioperative nurses. Perioperative nurses must be cognizant at all times of how long they are standing in one position or wearing too heavy of a weight. Specifically, they should not stand more than 30% of an 8 h workday or be in the same position for more than two hours, especially while wearing a lead apron. The AORN [26] offers four small changes that can be made for ergonomic safety: (1) the removal of unnecessary risks: and even small fixes can prevent one injury. (2) The application of formal fixes should be made, as quick fixes may cause more safety risks. (3) Safety data should be reviewed and clear plans made. Injury data should be combined with audit data so that action plans can be developed. (4) Safety rules should be rewritten using a three-column chart format. The first column outlines the safety action, the second column describes the potential risk of injury, and the third column describes actions for mitigating the risk.

There is a strong need for interventional research to reduce physical ergonomic demands on the perioperative team. Cha and colleagues [7] conducted simulation-based research utilizing exoskeletons on the perioperative team, including nurses. Exoskeletons are external devices that are worn to support physical demands and tasks and are currently being used in agriculture and manufacturing settings. This study’s findings demonstrate that perioperative team members were receptive to wearing an exoskeleton to decrease the risk of pain and injury; however, this technology is still in the development phase and is not ready for healthcare worker use.

In addition to exoskeletons, the use of body sensors, or wearable technology, also offers promise for ergonomic safety for perioperative nurses. Stefana and colleagues [27] conducted a systematic integrative review on the use of sensors for ergonomic safety. They identified 28 articles based on 24 studies, the majority of which propose wearing sensors to analyze unfavorable postures during working tasks, while others focused on sensors measuring physical loads. Most of the studies focused on construction workers, with only one study focused on healthcare workers. Unfortunately, as with exoskeletons, the use of sensors to improve ergonomic safety is still only being tested in lab studies and is not ready for the real-world context, and more research is needed.

Overlooking safety risk factors has significant implications for both the perioperative team, as well as the healthcare system. To date, while technology is being investigated to assist with worker ergonomics, awareness and prevention of safety risk factors are at the individual and organizational levels. Ignoring safety risk factors can lead to perioperative nurse injury, causing both short and long-term musculoskeletal pain. Worker injury costs the healthcare system through absenteeism, compensation claims, as well as decreased productivity.

## 6. Limitations

This study was conducted with perioperative nurses in only one large healthcare system. Only 120 different nurses in 60 different surgical procedures were observed; additional safety violations may have been missed. Additionally, while subjectivity may be a concern, the tool used to assess safety risks was developed from safety risks identified by the AORN and utilized in the operating room safety risk study by Yasak and Vural [8].

## 7. Conclusions

The risk factors of working in a perioperative setting are often overlooked in our fast-paced working environments. This study documents the specific safety risk factors perioperative nurses face in the operating room setting. Interventional research is urgently needed to reduce the safety of perioperative nurses if we are to retain a healthy, productive workforce to provide the highest quality of patient care.

## Figures and Tables

**Table 1 healthcare-11-00698-t001:** At risk safety issues by perioperative nurses.

Ergonomic Safety Risk	Circulating Nurse	Scrub Nurse	CRNA	RNFA	TOTAL
Awkward position	9	4	1	1	15
Forcible exertion	1	0	1	1	3
Lifting while bending or twisting	7	2	3	1	13
Overreaching	9	1	5	1	16
Repeated motions	0	0	0	0	0
Static posture	0	0	0	0	0
Wrist deviation	1	0	2	1	4
Lifting too-heavy objects	5	1	0	0	6
Pushing/pulling too much weight	1	0	0	0	1
Wearing a lead vest for too long	0	1	0	0	1
Wet floor	1	0	1	0	2
Opening too-heavy of a door	0	0	0	0	0
Transferring too-heavy of a patient/not enough help	4	1	2	0	7
Positioning too-heavy of a patient/not enough help	3	0	3	0	6
Standing in one position for too long	0	1	0	0	1
Carrying/pushing heavy oxygen tank	0	0	0	0	0
Debris on floor	5	2	0	0	7
Total number of at-risk behaviors	46	13	18	5	82

## Data Availability

Not applicable.

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
