# Peer review of "At Risk Safety Behaviors of the Perioperative Nursing Team: A Direct Observational Study"

_healthcare, 2023, doi:10.3390/healthcare11050698_

Round 1
Reviewer 1 Report
This article presents the results of a study on an interesting topic. Unfortunately, however, I have some major concerns, which mainly relate to:
The manuscript still needs extensive editinf of English.
The scope of the study is too narrow by examining a single measure without linking it in the context of other relevant measures to understand the findings and there is no theoretically based model.
The literature is not directly related to the research design. The article spent a great deal of time emphasizing the specific occupational hazards nurses face. Still, it did not touch the literature on what variables, based on previous studies, influence these hazards.
The main concern is about the method and analysis of results. Statistically the article is rather poor, more precise statistical analyses are needed to give more consistency to the findings found in their research whether quantitative, qualitative or mixed.
In the discussion, the authors need to explain their findings in more depth and part of the discussion is just a description of their results indicating that previous studies found the same relationships. The authors should broaden the implication of this research.
Limitations are not forcefully discussed and the authors should include a stronger call for future research.
The bibliographical references do not conform to the style required by the journal, so they should adapt them correctly.
Unfortunately, I am not convinced that this study makes a sufficiently broad contribution to scientific knowledge to justify its publication.
But I encourage the authors to refine their research article, as the topic is of great interest.
Author Response
Please see attachmebt

Reviewer 2 Report
I thank you for the opportunity to comment this study. The subject is very important.
Even I found this study interesting I think that it is impossible to get an overview how preoperative nurses work by analyzing only two nurses (even there were repeated analyses).
It remains also rather subjective whether all the findings really were risks for accident.
The conclusion, even quite long, does not manage to create very practical recommendations.
The reference list is chaotic because the numbering of references have failed.
I would like to recommend authors to submit this article as a letter or opinion.
Author Response
Dear Reviewer,
Thank you for your careful attention to our manuscript. See comments/changes below. We thank you for your time in strengthening this manuscript.
Comment: Even I found this study interesting I think that it is impossible to get an overview how preoperative nurses work by analyzing only two nurses (even there were repeated analyses).
Response: We are sorry this was not clear- we observed two nurses in 60 different operating rooms/procedures so there were 120 different nurses observed. Changes made.
Comment: It remains also rather subjective whether all the findings really were risks for accident.
Response: We understand your comment, however we utilized an observational check list that contained items the Association of periOperative Nurses (AORN) lists as safety risks as well as the items from a study of perioperative nurses that was conducted in Turkey (and is referenced in our paper). We added a limitation on subjectivity.
Comment: The conclusion, even quite long, does not manage to create very practical recommendations.
Response: We added more practical recommendations based specifically on the safety risks we identified.
Comment: The reference list is chaotic because the numbering of references have failed.
Response: We apologize for chaotic references. When our word document became a PDF the references became totally disorganized. This has been fixed.
Round 2
Reviewer 1 Report
First of all, I thank the authors for taking into account my recommendations for improvement. When reviewing the changes made by the authors, I consider that the article has improved considerably.
Therefore, the article meets all the characteristics to be published in its form.
Author Response
Thank you!
Reviewer 2 Report
Dear authors, thank you for corrections. I still find your study as a case study. It is impossible make these conclusions based on only two nurses even the repeated observations. But as a case study this work might be appropriate.
Author Response
Thank you for your second review. We are not sure why you still think we had only two nurses. We had 120 different nurses observed in 60 different operating room procedures. We made this more clear in the manuscript. .